# Glycerol contributes to tuberculosis susceptibility in male mice with type 2 diabetes

Nuria Martinez [1], Lorissa J. Smulan[1], Michael L. Jameson[1], Clare M. Smith [2], Kelly Cavallo[1], Michelle Bellerose [2], John Williams[2], Kim West[1], Christopher M. Sassetti[2], Amit Singhal[1,3,4,5,6] & Hardy Kornfeld [1,6] ✉

Diabetes mellitus increases risk for tuberculosis disease and adverse outcomes. Most people with both conditions have type 2 diabetes, but it is unknown if type 1 and type 2 diabetes have identical effects on tuberculosis susceptibility. Here we show that male mice receiving a high-fat diet and streptozotocin to model type 2 diabetes, have higher mortality, more lung pathology, and higher bacterial burden following *Mycobacterium tuberculosis* infection compared to mice treated with streptozotocin or high-fat diet alone. Type 2 diabetes model mice have elevated plasma glycerol, which is a preferred carbon source for *M. tuberculosis*. Infection studies with glycerol kinase mutant *M. tuberculosis* reveal that glycerol utilization contributes to the susceptibility of the type 2 diabetes mice. Hyperglycemia impairs protective immunity against *M. tuberculosis* in both forms of diabetes, but our data show that elevated glycerol contributes to an additional adverse effect uniquely relevant to type 2 diabetes.

Diabetes mellitus (DM) is a major driver of the global tuberculosis (TB) pandemic[1]. Poor glycemic control in diabetic patients increases TB risk[2] and is associated with greater TB severity, worse treatment outcomes, and increased recurrence[3]. We previously reported that mice with streptozotocin (STZ)-induced pancreatic beta cell death, insulin deficiency, and chronic hyperglycemia as occurs in type 1 DM, suffer increased TB severity with higher lung bacterial load and more lung immune pathology than euglycemic mice[4]. Delayed adaptive immune priming is the major susceptibility mechanism in STZ-treated mice[5]. This results from defective innate recognition of inhaled *M. tuberculosis* (*Mtb*) by resident alveolar macrophages due to reduced expression of MARCO and CD14 that are co-receptors with TLR2 for the *Mtb* cell wall component trehalose dimycolate[6]. We also reported that naive T cells from hyperglycemic mice have a hyperactive proliferation and cytokine response to T cell receptor stimulation, which along with higher bacterial load might

further contribute to increased immune pathology and delayed resolution[7].

Both type 1 and type 2 DM impair TB defense, but over 95% of people living with TB-DM comorbidity have type 2 DM[8] that features dyslipidemia and glucotoxicity[9]. In people with comorbid type 2 DM and TB, changes in lipid metabolism might further exacerbate the immunopathy that results from chronic hyperglycemia alone[10]. The pathogenesis of type 2 DM is complex and has multiple genetic associations. It begins with visceral adiposity and insulin resistance, and further progression is linked with adipose tissue inflammation. Full expression of type 2 DM occurs when beta cell loss results in relative and finally absolute insulin deficiency[11]. Polygenic mouse type 2 DM models are challenging for TB research due to variable penetrance, moderate hyperglycemia, and issues of genetic background controls[12].

Common inbred strains of mice on a high fat diet (HFD) develop insulin resistance, but do not exhibit beta cell loss or severe

[1]Department of Medicine, University of Massachusetts Chan Medical School, Worcester, MA, USA. [2]Department of Microbiology and Physiological Systems, University of Massachusetts Chan Medical School, Worcester, MA, USA. [3]A*STAR Infectious Diseases Labs (ID Labs), Agency for Science, Technology and Research (A*STAR), Singapore 138648, Singapore. [4]Singapore Immunology Network (SIgN), Agency for Science, Technology and Research (A*STAR), Singapore 138648, Singapore. [5]Lee Kong Chian School of Medicine, Nanyang Technological University, Singapore 636921, Singapore. [6]These authors contributed equally: Amit Singhal, Hardy Kornfeld. ✉e-mail: Hardy.Kornfeld@umassmed.edu

hyperglycemia as occurs with human type 2 DM in TB-endemic regions[13]. We hypothesized that the combined perturbation of glucose and lipid metabolism characteristic of type 2 DM, has additive or synergistic effects on TB severity as reported with microvascular complications[14]. To test that hypothesis, we put mice on HFD and then treated with STZ (HFD-STZ) to mimic the accelerated beta cell loss found in human type 2 DM[15,16]. We found that HFD-STZ mice had hyperglycemia, elevated total and LDL cholesterol, glucose intolerance, insulin resistance, and they did not produce insulin. They suffered 50% mortality by 11 weeks after aerosol *Mtb* infection and had higher lung bacterial burden, and more lung immune pathology compared to mice on HFD or STZ alone. We identified deficits in innate and adaptive immune responses to infection like those previously reported in the chronic STZ type 1 DM mouse model[5–7], but we also discovered a susceptibility mechanism unique to type 2 DM. HFD-STZ mice exhibited adipose tissue wasting accompanied by elevated circulating free fatty acids and glycerol. A mutant strain of *Mtb* unable to use glycerol as a carbon source (H37Rv Δ*glpK*) had reduced virulence compared to the parental H37Rv in the type 2 DM mouse model but not in euglycemic control mice, or in mice treated with STZ or fed HFD alone.

## Results

### HFD-STZ mice exhibit hypoinsulinemia, insulin resistance, hyperglycemia, and dyslipidemia

C57BL/6 J mice were fed a standard chow diet (CD) or HFD and after 4 weeks on diet were treated or not with STZ (Fig. 1a). After 8 weeks on diet and 4 weeks post-STZ treatment, mice were infected with ~100 colony forming units (CFU) of *Mtb* Erdman by aerosol. Uninfected and infected mice on HFD alone gained more weight than those on CD, while STZ and HFD-STZ mice had body weight trends comparable to CD mice (Fig. 1b and S1a). Regardless of the condition or treatment, body weight reached a plateau after infection with *Mtb*, with a trend for higher body weight in HFD mice (Fig. 1b). In contrast, uninfected HFD mice continued gaining weight until reaching an average of ~42 g that was significantly higher than uninfected mice in other groups (Fig. S1a). At the time of infection, HFD mice had higher body weight than the other groups (Fig. 1c). Infected and uninfected mice on HFD had comparable blood glucose levels as mice on CD for the duration of the experiment (Fig. 1d and S1b). As expected, STZ treatment alone or with HFD resulted in elevated blood glucose levels. At the time of the infection, HFD-STZ mice had significantly higher levels of blood glucose than the other groups (Fig. 1e). Plasma insulin levels measured in uninfected (Fig. S1c) and infected mice (Fig. 1f) were significantly higher in HFD mice compared to the other groups. Glucose tolerance tests (GTT) and insulin tolerance tests (ITT) were performed in all the groups two weeks before harvest (6 weeks post infection [p.i.]) and in uninfected mice (Fig. 1g, h and S1d, e). In the absence of infection, HFD mice were glucose intolerant compared to CD (Fig. S1d), but this difference was not significant in infected mice (Fig. 1g). No difference in glucose tolerance was seen between HFD-STZ and STZ mice. Mice in all groups responded to exogenous glucose and restored levels back to the baseline value. The ITT showed that HFD-STZ mice had a delayed response to insulin in both uninfected and infected conditions (Fig. 1h and S1e). The area under the curve (AUC) was increased in the HFD-STZ mice, but only in the first 30 min of the assay and while statistically significant, the biological relevance is uncertain. In addition to hyperglycemia and insulin resistance, uninfected HFD-STZ mice exhibited dyslipidemia with elevated plasma cholesterol levels (Fig. S1f). After 8 weeks of infection only HDL levels were maintained high in HFD-STZ mice, compared to control mice.

### Combined HFD and STZ increases TB severity

Mice that received HFD-STZ treatment prior to *Mtb* challenge had significantly reduced survival compared to HFD or STZ alone, with

~50% mortality by 11 weeks p.i. (Fig. 2a). Survival was similar for all the groups in uninfected conditions (Fig. S2a). The infected HFD-STZ mice also had the highest lung CFU and lesion area at 8 weeks p.i. (Fig. 2b, c). Mice treated with STZ alone (4 weeks of pre-infection hyperglycemia) had higher lung CFU than CD mice, but they did not exhibit decreased survival or severe lung pathology. Lung CFU at 4 and 20 weeks p.i. was also higher in HFD-STZ mice than the other groups (Fig. S2b), but there were no differences between groups in spleen or liver CFU (Fig. S2c).

Lung leukocytes from infected HFD-STZ mice showed an increased percentage and absolute number of CD11c⁺CD11b⁻Ly6C^lo alveolar macrophages (AMs) and CD11c⁺CD11b⁺Ly6C^lo MHCII⁺ dendritic cells (DCs) and activated macrophages (Fig. 2d), calculated with the total number of live cells in the lung (Fig. S2d) and compared to the other groups of mice at 8 weeks p.i. (gating strategy in Fig. S2e). The percentage and number of CD3⁺ cells were significantly higher in the HFD and HFD-STZ mice 8 weeks p.i. There was no difference in the percentage or number of CD11c⁻CD11b⁺Ly6G⁺ neutrophils between groups. Levels of selected cytokines were measured in whole lung homogenates by ELISA. No significant differences in IFN-γ, TNF-α or IL-6 were identified between groups (Fig. 3a). Notably, IL-1α levels were significantly higher in HFD and HFD-STZ mice compared to CD, while IL-1β was uniquely elevated in the lungs of HFD-STZ mice (Fig. 3a). IL-1α has been associated with *Mtb* virulence and macrophage necrosis[17], and IL-1β can be both a mediator of antimycobacterial immunity and damaging immune pathology during *Mtb* infection[18]. Moreover, foam cells are associated with chronic inflammation and may contribute to inflammation and cell death in TB[19]. Accordingly, we found increased terminal deoxynucleotidyl transferase dUTP nick end labeling (TUNEL) and oil red O staining in the lungs of HFD-STZ mice (Fig. 3b, c and Fig. S2f). After in vitro infection with *Mtb* Erdman MOI 10, bronchoalveolar lavage (BAL) cells from HFD-STZ mice were bigger, compared to control, probably due to a higher content of lipids, observed with oil red O staining (Fig. S2f, g). Taken together, the histopathology, flow cytometry, and cytokine results supported the conclusion that HFD-STZ mice are prone to more severe TB immune pathology than mice on HFD or mice having 4 weeks of STZ-induced hyperglycemia alone.

### Innate and adaptive immune perturbation in HFD-STZ mice

We previously reported that STZ-treated mice with chronic (≥ 8 weeks) hyperglycemia mount a delayed innate immune response to inhaled *Mtb*, which bridges to a delayed but hyperactive cell-mediated immune response[4,5]. We also showed that AMs from chronically hyperglycemic mice have impaired recognition and phagocytosis of *Mtb* due to reduced expression of MARCO and CD14 that are co-receptors for the *Mtb* cell wall lipid trehalose dimycolate[6]. These findings support a model where a sluggish innate response to initial infection slows delivery of *Mtb* antigen to the lung-draining lymph nodes, thereby delaying T cell priming at a time when bacterial burden is rising exponentially. We further showed that T cells from chronically hyperglycemic mice proliferate more and produce more Th1, Th2, and Th17 cytokines in response to T cell receptor stimulation than cells from euglycemic mice[7]. This hyperreactivity, along with higher bacterial load resulting from delayed T cell priming, might contribute to the increased TB immune pathology in diabetic mice. Therefore, we next investigated the mechanism(s) exacerbating immune pathology in *Mtb*-infected HFD-STZ mice. We observed reduced gene expression of *MARCO* and *CD14* in BAL cells of uninfected HFD-STZ mice compared to CD mice, although only the difference in *CD14* reached statistical significance (Fig. S3a). Lung-draining lymph nodes from HFD-STZ mice contained a significantly lower percentage of CD4⁺IFN-γ⁺ and CD8⁺IFN-γ⁺ T cells compared to CD mice at 14 days p.i. (Fig. S3b), while the percentage of TNF-α⁺ T cells was similar between groups (Fig. S3c). No differences in IFN-γ⁺ or TNF-α⁺ T cells were seen in the lung (Fig. S3d, e). HFD-STZ mice also had a lower percentage of activated CD4⁺CD69⁺ T cells in lung-draining lymph nodes (Fig. S4a) and lower

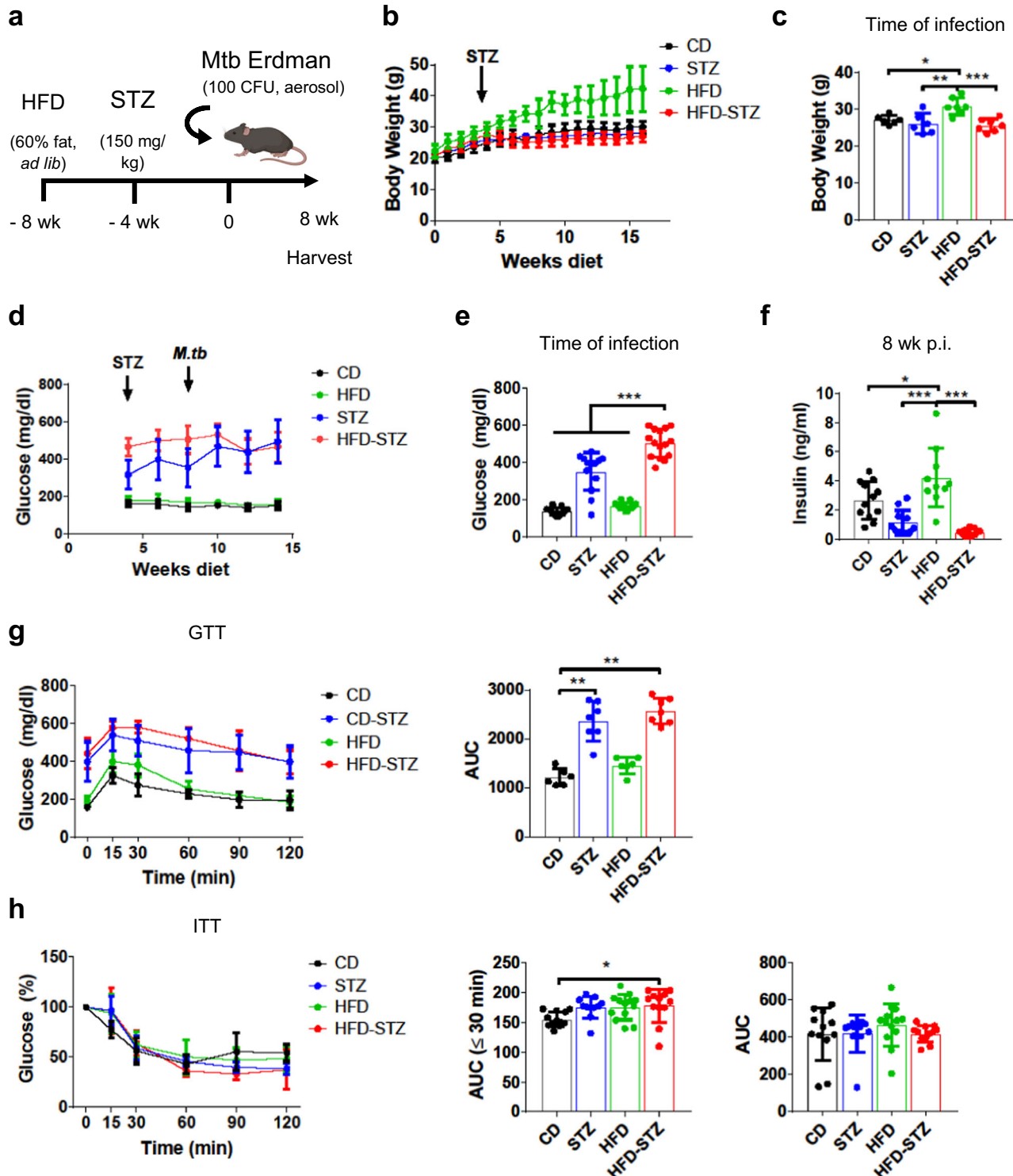

**Fig. 1 | Streptozotocin combined with high fat diet mirrors human type 2 diabetes. a** Mice were injected with streptozotocin (*STZ*, 150 mg/kg) 4 weeks after starting a high fat diet (*HFD*, 60% kcal % fat) and infected with ~100 CFU *Mtb* Erdman by aerosol 4 weeks after STZ treatment. Comparison was made with mice fed HFD or control diet (*CD*) without STZ treatment, and mice on control diet treated with STZ. Graphic created with BioRender.com. **b** Body weight was measured weekly during the experiment (*n* = 6 mice). **c** Body weight at the time of infection (*n* = 6 mice). **d** Non-fasting blood glucose was quantified biweekly (*n* = 15 mice). **e** Non-fasting blood glucose levels at the time of infection (*n* = 13 mice for STZ and *n* = 15 mice for the others). **f** Insulin levels in plasma 8 weeks p.i. (*n* = 12 mice for CD, *n* = 11

mice for STZ and HFD-STZ and *n* = 10 mice for HFD) **g** Glucose tolerance test (*GTT*) (*n* = 6 mice for CD and HFD and *n* = 7 mice for STZ and HFD-STZ) and (**h**) insulin tolerance test (*ITT*) (*n* = 13 mice for HFD, *n* = 12 mice for CD and HFD-STZ and *n* = 11 mice for STZ) were performed 6 weeks p.i. Area under the curve (*AUC*) was calculated for the first 30 min for ITT (*middle*) or for the total for both GTT and ITT (*right*). Data are expressed as mean ± SD. The experiments were repeated at least twice. Statistical analysis was performed by One-Way ANOVA, *$P < 0.05$, **$P < 0.01$ and ***$P < 0.001$ (*p*-value from left to right = (**c**): 0.0210, 0.0026, 0.0007; (**e**): all < 0.0001; (**f**): 0.0305, < 0.0001, < 0.0001; (**g**): all < 0.0001; (**h**): 0.0489). Source data are provided as a Source Data file.

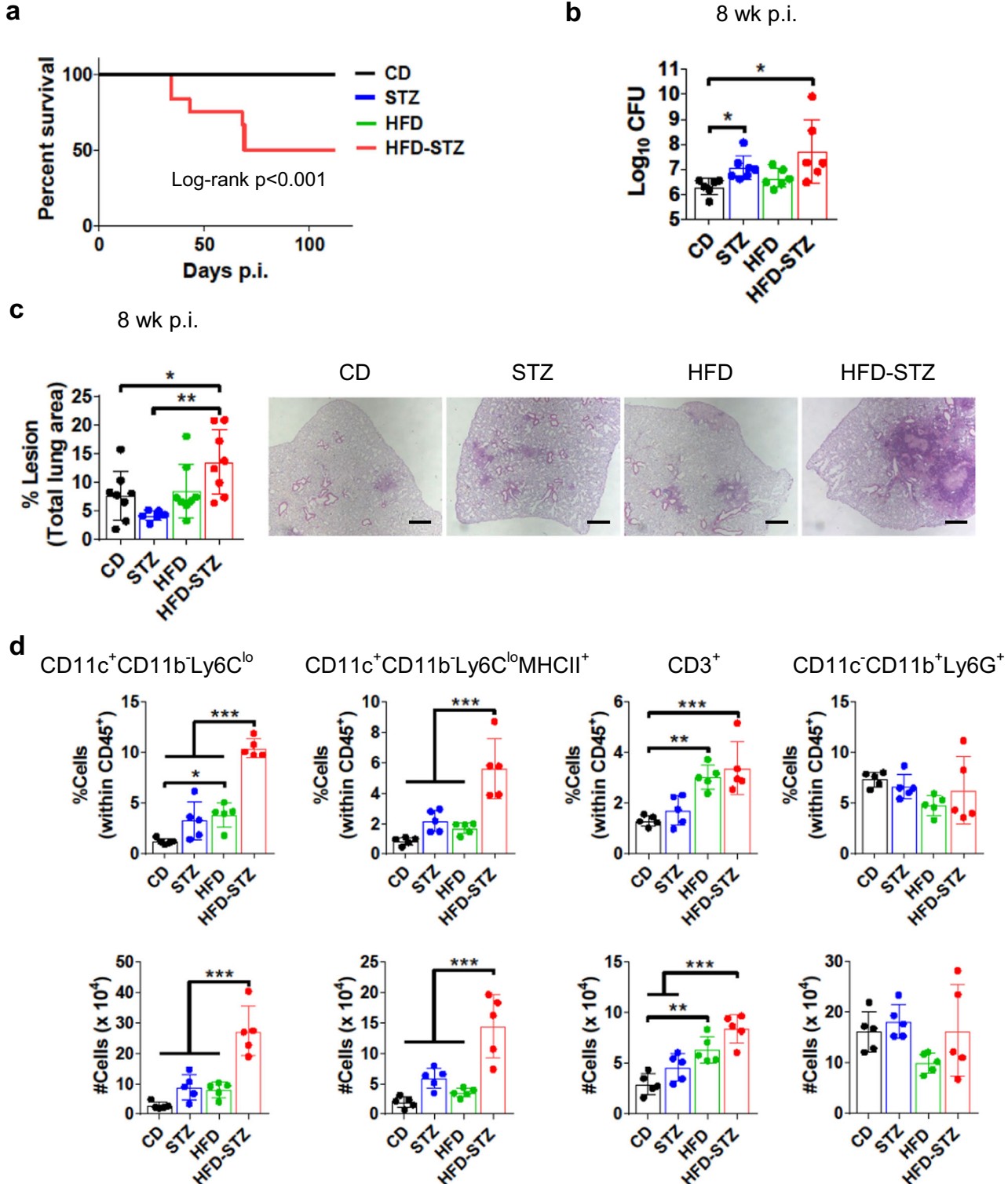

**Fig. 2 | Combined dyslipidemia and hyperglycemia increase TB severity.**
Untreated mice (*CD*) or mice treated with a high fat diet (*HFD*) or streptozotocin
(*STZ*) or a combination of both (*HFD-STZ*) were aerosol infected with *Mtb* Erdman
(∼100 CFU) and samples and tissues were taken 8 weeks p.i. **a** Survival plot (*n* = 12).
Analysis was performed with log-rank Mantel-Cox test. **b** Lung bacterial load (*n* = 6
mice for CD, HFD and HFD-STZ and *n* = 7 mice for STZ). **c** Lung lesion area
expressed as a percentage of total lung area measured (*left*) and representative lung
sections (*right*) from mice in each group (*n* = 6 mice for STZ and *n* = 8 mice for the
others). Scale bar, 500 μm. **d** Flow cytometry of lung leukocytes with mAb staining
for the indicated populations (*n* = 6 mice). The percentage of cells (within CD45+

cells) and the total number of cells back calculated to the total lung cell numbers
are presented in the upper and lower graphs, respectively. All data are expressed as
mean ± SD. The experiments were repeated at least three times. Statistical analysis
was performed by One-Way ANOVA, *$P$ < 0.05, **$P$ < 0.01, and ***$P$ < 0.001 (*p*-value
from left to right = (**b**): 0.0431, 0.0094; (**c**): 0.0342, 0.0029; (**d**): top left 0.0173,
< 0.0001, < 0.0001, < 0.0001, top second < 0.0001, 0.0006, 0.0001, top third
0.0037, 0.0007, bottom left all < 0.0001, bottom second < 0.0001, 0.0010,
< 0.0001, bottom third < 0.0001, 0.0014, 0.0035). Source data are provided as a
Source Data file.

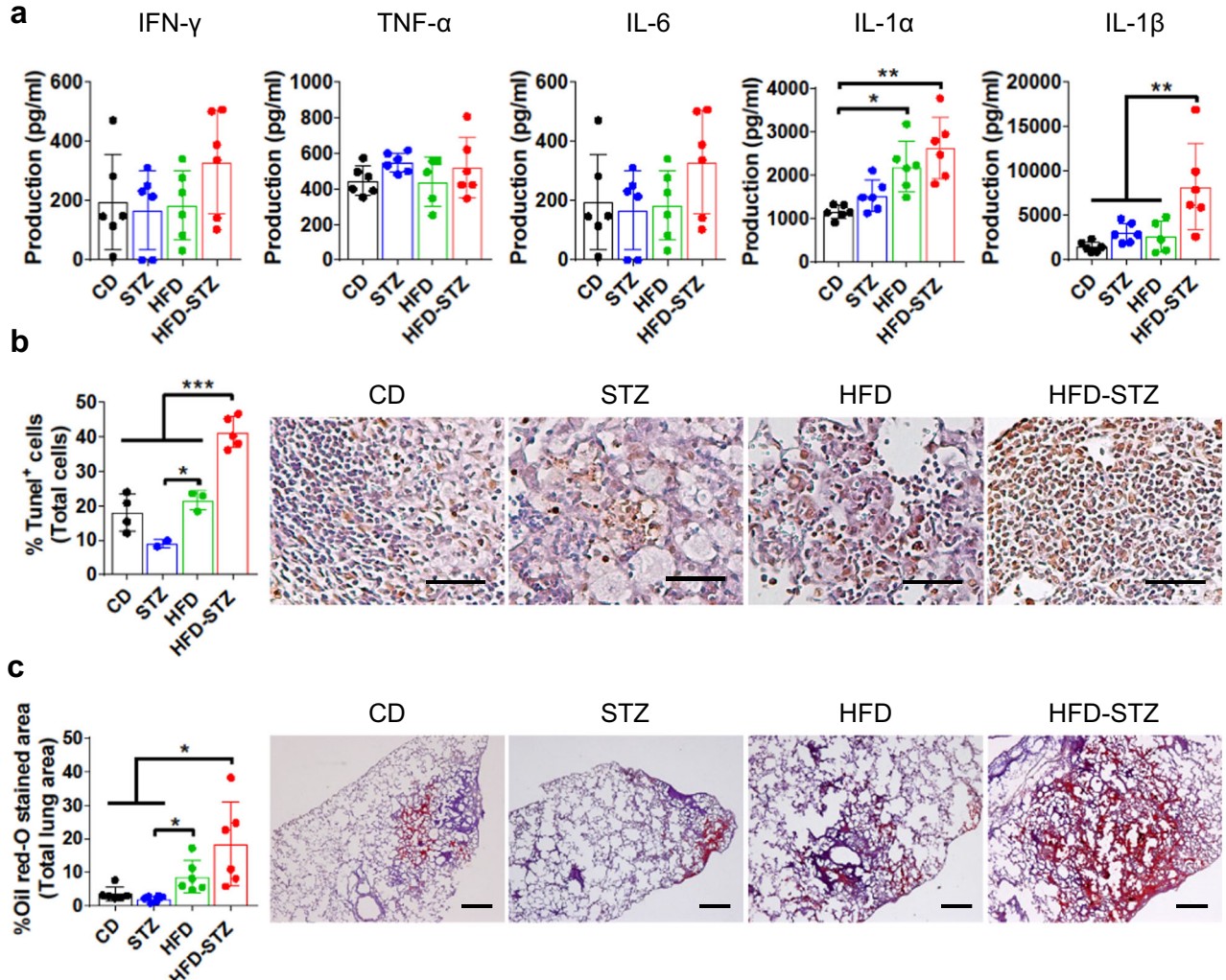

**Fig. 3 | Combined dyslipidemia and hyperglycemia increase cell death and lipid accumulation in the lung.** Untreated mice (*CD*) or mice treated with a high fat diet (*HFD*) or streptozotocin (*STZ*) or a combination of both (*HFD-STZ*) were aerosol infected with *Mtb* Erdman ( ~ 100 CFU) and samples and tissues were taken 8 weeks p.i. **a** IFN-γ, TNF-α, IL−6, IL-1α and IL-1β in lung homogenates were quantified by ELISA (*n* = 6 mice). **b** Percentage of terminal deoxynucleotidyl transferase dUTP nick end+ cells (*left*) and representative lung sections from each group (*right*) (*n* = 5 mice for HFD-STZ and *n* = 4 mice for the rest). Brown color shows positive signal.

Scale bar, 500 μm. **c** Oil Red O staining of lung sections from infected CD, STZ, HFD and HFD-STZ mice. Percentage of oil red-O positive area, in proportion of total lung area (*left*) (*n* = 6 mice). Representative lung sections from each group (*right*). Scale bar, 500 μm. All data are expressed as mean ± SD. The experiments were repeated at least three times. Statistical analysis was performed by One-Way ANOVA, *$P < 0.05$ and **$P < 0.01$ (*p*-value from left to right = (**a**): IL-1α 0.0151, 0.0017 IL-1β 0.0020, 0.0145, 0.0176; (**b**): < 0.0001, < 0.0001, 0.0396, 0.0004; (**c**): 0.0423, 0.0256, 0.0010). Source data are provided as a Source Data file.

percentage of CD8+CD69+ T cells in lungs (Fig. S4b) compared to CD mice. These results were consistent with our earlier studies linking reduced bacterial recognition by AMs from mice with chronic STZ-induced hyperglycemia to delayed T cell priming in the lung-draining lymph nodes and late control of *Mtb* replication in the lung[5,6].

We attributed the non-antigen specific T cell hyperreactivity of chronically hyperglycemic mice to a receptor for advanced glycation endproducts (RAGE)-dependent pre-activated functional state featuring nuclear chromatin decondensation in naïve T cells[7]. To assess this phenotype in the type 2 DM model, the nuclei cells from uninfected CD or HFD-STZ mice were measured (Fig. S4c). Cells were distributed into bins by nucleus diameter and those having nuclei > 6 μm in diameter were classified as pre-activated. The percentage of pre-activated T cells from uninfected HFD-STZ mice was significantly higher than from CD mice (Fig. S4c, middle graph and image). Thus, the impact of HFD combined with 4 weeks of STZ-induced hyperglycemia mirrored the characteristics we previously described in mice on a standard diet with > 8 weeks of hyperglycemia[7]. Overall, the TB susceptibility phenotype

and immune function perturbations in HFD-STZ type 2 DM model identified an additive or synergistic detrimental effect of hyperglycemia and altered lipid metabolism on host defense, reminiscent of their combined effect on the vascular complications of type 2 DM[14,20].

## Combined HFD and STZ induces adipose tissue lipolysis and hyperlipidemia

Adipose tissues store energy from food as lipids and are the only tissues that accommodate abundant food intake by increasing cell size (hypertrophy) or producing new cells from resident stem cells (hyperplasia). Overnutrition triggers systemic metabolic imbalance and defective lipid storage. While mice on HFD for 14 weeks had enlarged perigonadal white adipose tissue (pgWAT), STZ treatment abrogated that process (Fig. 4a). Liver mass was similar between all the groups, but skeletal muscle mass was significantly decreased in STZ-treated mice before and after infection (Fig. S5a). Adipocyte cell size in pgWAT of *Mtb*-infected HFD mice or age-matched uninfected HFD mice was higher than all other groups (Fig. 4b and S5b). In contrast,

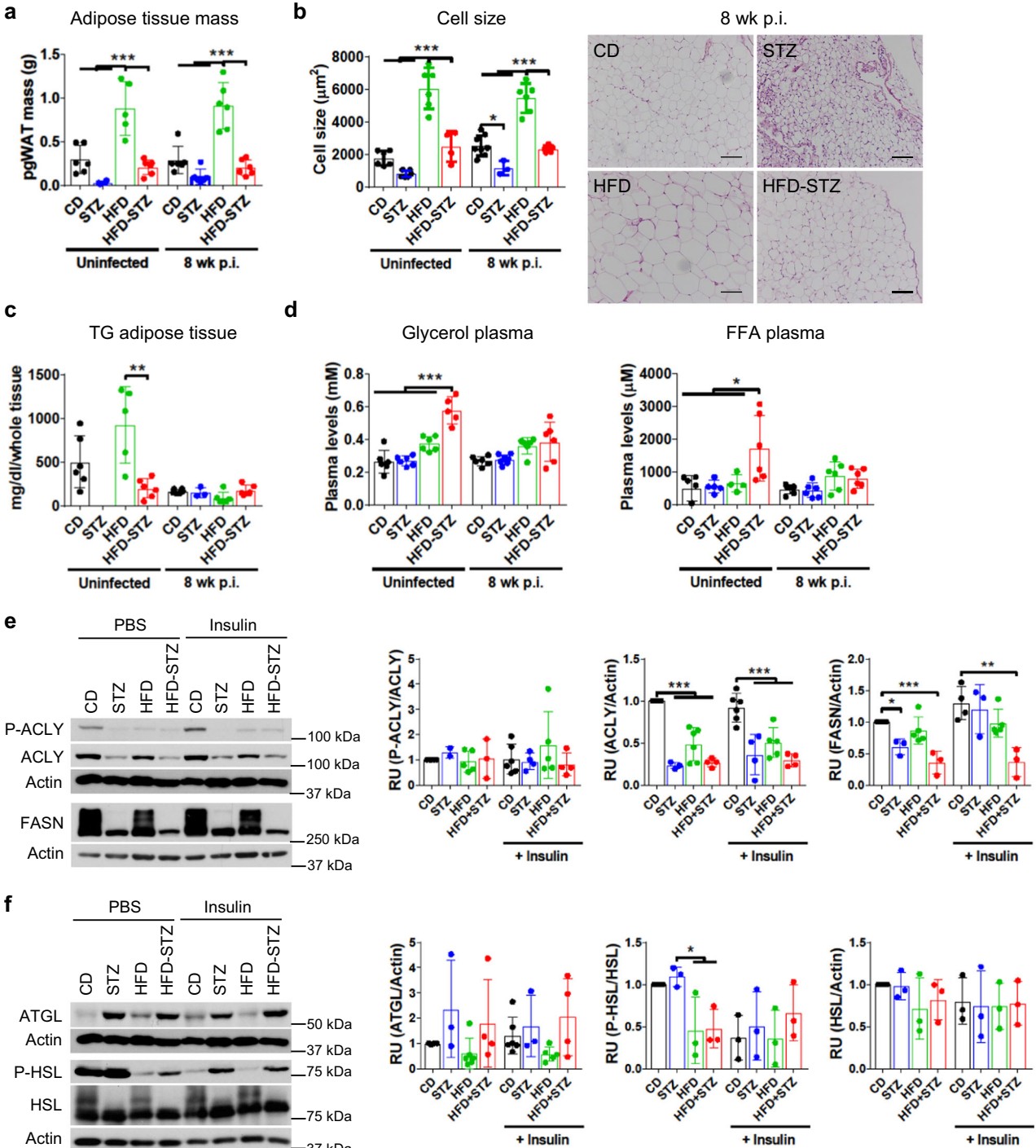

**Fig. 4 | Combination of high fat diet with streptozotocin treatment induces adipose tissue loss.** Control diet mice (*CD*), mice treated with streptozotocin (*STZ*), mice fed high fat diet (*HFD*) and mice fed high fat diet and treated with streptozotocin (*HFD-STZ*) infected with -100 CFU *Mtb* Erdman for 8 weeks were compared to uninfected age-matched mice. **a** Mass of perigonadal white adipose tissue (*pgWAT*) (*n* = 4 mice for STZ, *n* = 5 mice for HFD and *n* = 6 mice for the others). **b** Average cell size of adipocytes in pgWAT (*left*) (*n* = 4 mice for STZ and HFD-STZ and *n* = 6 mice for the others) and representative images of H&E-stained pgWAT of infected mice (*right*). Scale bar, 100 μm. **c** Content of TG in pgWAT (*n* = 3 mice for STZ and *n* = 6 mice for the others). **d** Plasma levels of glycerol (*n* = 6 mice) and free fatty acids (*FFA*) (*n* = 5 mice for STZ and HFD, *n* = 6 mice for the others). Uninfected CD, HFD, STZ or HFD-STZ mice were intraperitoneally injected with PBS or 1 mg/kg insulin and after 15 min blood and pgWAT were harvested. **e** De novo

lipogenesis was analyzed by measuring P-ACLY, total ACLY and FASN (*n* = 3 mice for STZ and HFD-STZ, *n* = 4 mice for the others). **f** Lipolysis was analyzed by measuring ATGL, P-HSL and total HSL (*n* = 3 mice for STZ and HFD-STZ, *n* = 4 mice for the others). Equal number of samples from each group were run on each gel/blot and they were run in parallel in the case of ATGL. Differences were expressed as fold change to the CD group to compare between different blots and using β-actin as loading control that was probed in the same gel or run in parallel in the case of ATGL. Data are expressed as mean ± SD. Statistical analysis was performed by One-Way ANOVA for uninfected, 8 weeks p.i., untreated or insulin separately, *$P < 0.05$, **$P < 0.01$ and ***$P < 0.001$ (*p* value from left to right = (**a**): all < 0.0001; (**b**): all uninfected < 0.0001 infected 0.0247 and 0.0001 the rest; (**c**): 0.0034; (**d**): left all < 0.0001, right 0.0127, 0.0236, 0.0588; (**e**): middle all < 0.0001, right 0.0190, 0.0005, 0.0040; (**f**): 0.0421, 0.0497). Source data are provided as a Source Data file.

STZ treatment was associated with significantly smaller adipocyte size in both control and HFD mice.

Lipid storage is regulated by de novo lipogenesis from glucose while adipose tissue loss results from lipolysis that converts stored triglycerides (TG) into glycerol and free fatty acids (FFA) which are released to circulation[21]. Lipogenesis is induced by insulin-Akt signaling and the key enzymes are ATP citrate lyase (ACLY), acetyl-CoA carboxylase (ACC), and fatty acid synthase (FASN). Lipolysis is mediated by adipose triglyceride lipase (ATGL), hormone sensitive lipase (HSL), and monoglyceride lipase (MGL). This process is regulated by hormones like insulin or catecholamines in response to nutrient intake and energy demand[21,22]. Mice treated with STZ are insulin deficient, have downregulated de novo lipogenesis and upregulated lipolysis[23]. To investigate lipid metabolism during TB, we measured TG in pgWAT from uninfected mice or 8 weeks after aerosol Mtb infection (Fig. 4c). Uninfected HFD-STZ mice had significantly less TG in whole pgWAT compared to CD and HFD mice. At 8 weeks p.i. all groups of mice had comparable TG content, reflecting a reduction from the uninfected baseline in CD and HFD mice. We next quantified the products of lipolysis, namely glycerol and FFA, in plasma (Fig. 4d). Uninfected HFD-STZ mice had significantly higher glycerol and FFA levels than the other groups of mice, but this was abrogated by 8 weeks postinfection.

We evaluated de novo lipogenesis and lipolysis in pgWAT from uninfected mice injected with PBS or insulin 15 min prior to harvest by measuring the expression of relevant enzymes (Fig. 4e, f). Total ACLY and FASN protein levels were decreased in the STZ and HFD-STZ mice, indicating that de novo lipogenesis was downregulated by insulin deficiency. ATGL was increased in STZ and HFD-STZ mice, compared to CD or HFD mice, whether treated with PBS or insulin. Activated HSL (phospho-HSL) was higher in CD and STZ mice compared to HFD and HFD-STZ treated with PBS, while insulin treatment reduced phospho-HSL in CD and STZ mice, but not in HFD or HFD-STZ mice, confirming that HFD mice were insulin-resistant. These results indicated that dietary glucose and lipids were not being stored in the adipose tissue and that the products of lipolysis were increased in the HFD-STZ mice before infection. The white adipose tissue wasting in HFD-STZ mice as compared to HFD mice (Fig. 4a, b) was therefore attributable to decreased de novo lipogenesis and increased lipolysis with leakage of glycerol and FFA.

### Glycerol contributes to the TB severity in HFD-STZ mice

Host lipids represent a major carbon source for Mtb growth and survival under immune pressure in vivo[24,25], but glycerol is the preferred carbon source for Mtb in broth culture[26]. To test the effect of circulating glycerol on Mtb growth and host susceptibility in the mouse type 2 DM model, we used a glycerol kinase (glpK) deletion mutant of Mtb H37Rv (ΔglpK)[27] (Fig. 5a). This mutant was unable to catabolize glycerol and therefore cannot use glycerol as a carbon source, but this does not affect its growth or persistence in the lungs of metabolically normal C57BL/6J mice[27,28]. We infected control, STZ, HFD, and HFD-STZ mice with the parental H37Rv or with the ΔglpK mutant (Fig. 5a). Mice in both the STZ and HFD-STZ groups infected for 24 weeks with H37Rv had higher lung CFU than the control or HFD groups, but following infection with ΔglpK, bacterial burden was only increased in STZ mice (Fig. 5b). Lung lesion area was significantly greater only in HFD-STZ mice infected with H37Rv, but this difference to the other groups was not seen after infection with the ΔglpK (Fig. 5c). H37Rv infected HFD-STZ mice also showed 33% mortality by 154 days p.i. with no further mortality to the 168-day (24 weeks) scheduled end point (Fig. S6a). In contrast, 100% of HFD-STZ mice infected with ΔglpK survived by 24 weeks. In this experiment, HFD-STZ mice infected with H37Rv progressively lost weight over time, whereas body weight remained stable in HFD-STZ mice infected with the ΔglpK mutant (Fig. S6b). Complementation of ΔglpK with Mtb glpK inserted at the L5 attP site

(Glpk-Comp strain) restored the elevated lung CFU and pathology in HFD-STZ mice that was seen with parental H37Rv infection (Fig. 5d, e). Cumulatively, these data indicated that glycerol may contribute to increased TB severity in HFD-STZ mouse model of type 2 DM.

We then measured plasma glycerol levels in mice infected with H37Rv or ΔglpK strain at several time points p.i. (Fig. S6c). In mice infected with H37Rv or ΔglpK mutant glycerol levels remain elevated in HFD-STZ mice until at least 8 weeks p.i., confirming that differences in glycerol levels did not confound interpretation of the ΔglpK strain's TB phenotype in HFD-STZ mice. To further test the hypothesis that elevated glycerol exacerbates TB severity in HFD-STZ mice, we fed CD and STZ mice with a 5% glycerol solution in water ad libitum 8 weeks before infection with Mtb Erdman (Fig. 5g). This dose of glycerol achieved higher levels of glycerol in plasma of treated compared to untreated mice (Fig. S6d). Lung bacterial burden was significantly increased only in the mice treated with glycerol water and STZ (GW-STZ), compared to the other groups of mice (Fig. 5h). However, the lesion areas were just slightly increased in the GW-STZ, being only significant when compared to the STZ mice (Fig. 5i). Taken together, these results suggest that a combination of hypoinsulinemic hyperglycemia (STZ treatment) with high glycerol possibly results in increased TB severity.

## Discussion

The TB susceptibility of people living with DM has been extensively studied in the last decade[29–33], but the underlying mechanisms remain incompletely understood. We previously reported increased TB severity in mice with sustained hyperglycemia after a single dose of STZ[4]. We chose that DM model because chronic hyperglycemia is the primary driver of diabetic complications[34]. The full TB susceptibly phenotype required ≥ 8 weeks of hyperglycemia, at which time glycohemoglobin levels were also increased[4]. This fits with our concept of diabetic immunopathy as a complication similar to the other end organ damage in DM[31]. Using STZ avoids the confounding effects of autoimmunity, complex host genetics, and moderate hyperglycemia that complicate the use of other mouse DM models for TB research. There is, however, a strong rationale to model type 2 DM since it predominates in populations with high TB burden. We speculated that the dyslipidemia associated with type 2 DM could further exacerbate the immunopathy caused by hyperglycemia alone. That notion was reinforced by initial evidence from the present study showing that these combined metabolic perturbations synergistically increased TB severity (Fig. 2).

High fat diet promotes insulin resistance in mice but does not trigger pancreatic beta cell death or reduced insulin production[12]. Insulin resistance with compensatory increase in insulin production precedes human type 2 DM. This is followed by declining pancreatic beta cell function that is reduced by roughly 50% when type 2 DM is typically diagnosed, with additional loss of roughly 5% annually[35]. To model the beta cell loss seen in human type 2 DM, we combined STZ with HFD. This led to insulin resistance combined with hypoinsulinemia, loss of adipose tissue mass and body weight, elevated cholesterol, and increased circulating glycerol and FFA (Fig. 4). In the current study, we observed that the combination of HFD and acute STZ-induced hyperglycemia reduced survival and increased immune pathology after aerosol Mtb challenge in contrast to either HFD or STZ alone. This synergism between disordered glucose and lipid metabolism has not previously been described in the context of TB but is a known factor in other diabetic complications[36–38].

Alternative type 2 DM models used for TB research include the combined STZ plus nicotinamide (NA), or an energy-dense diet (EDD) without drug treatment[39–41]. These models along with HFD-STZ have identified TB phenotypes potentially relevant to human comorbidity, but each has limitations. Cheekatla et al.[39] reported that NK and CD11c+ cell interactions drive IL-6 production and increased immune pathology with shortened survival in STZ-NA mice infected by aerosol with Mtb H37Rv. The IL-6 elevation and lung CFU difference of STZ-NA vs

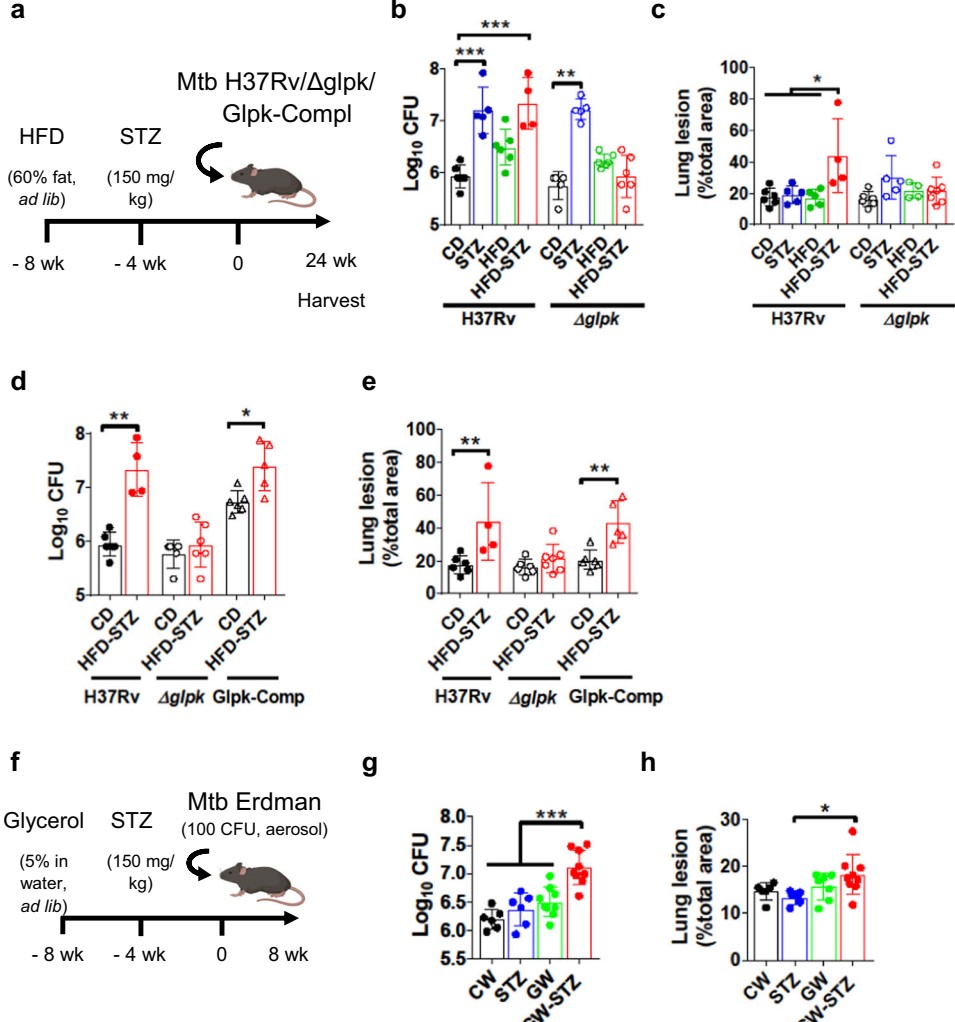

**Fig. 5 | Mutation of bacterial glycerol-3-kinase abrogates the enhanced virulence of *M. tuberculosis* in diabetic mice. a** Untreated mice (*CD*) or mice treated with a high fat diet (*HFD*) or streptozotocin (*STZ*) or a combination of both (*HFD-STZ*) were aerosol infected with wild type H37Rv, H37Rv lacking glycerol-3-kinase (*ΔglpK*) or H37Rv glpk complement (*Glpk-Compl*) (~100 CFU) for 24 weeks. Graphic created with BioRender.com. **b** Lung bacterial burden (*n* = 5 mice for STZ and HFD-STZ, *n* = 6 mice for the others). **c** Lung lesion area expressed as percentage of total lung area (*n* = 5 mice for STZ and HFD-STZ, *n* = 6 mice for the others). **d** Lung bacterial burden of CD and HFD-STZ mice infected with H37Rv, Δglpk and glpk-compl (*n* = 6 mice). **e** Lung lesion area expressed as percentage of total lung area (*n* = 6 mice). **f** Mice were left untreated (*CW*), treated with 5% glycerol water (*GW*) ad libitum 8 weeks before infection or with STZ for 4 weeks before infection (*STZ*) or a

combination of both (*GW-STZ*). All groups of mice were aerosol infected with Mtb Erdman (~100 CFU), treatment of glycerol water was maintained for the duration of the experiment and tissues were harvested 8 weeks p.i. (*n* = 6 mice for CW and STZ and *n* = 9 mice for the others). Graphic created with BioRender.com. **g** Lung bacterial burden **h** Lung lesion area expressed as percentage of total lung area. Data are expressed as mean ± SD. The experiments were repeated at least twice. Statistical analysis was performed for each strain by One-Way ANOVA in (**b, c, h, i**) or two-tailed Student's t-test for (**d, e**), *$P < 0.05$, **$P < 0.01$ and ***$P < 0.001$ (p-value from left to right = (**b**): 0.0002, 0.0001, 0.0048; (**c**): 0.0114, 0.0226, 0.0129; (**d**): 0.0095, 0.0173; (**e**): 0.0095, 0.0087; (**g**): < 0.0001, 0.0003, < 0.0001; (**h**): 0.0261). Source data are provided as a Source Data file.

euglycemic mice did not emerge until 6 months p.i. Infected STZ-NA mice had 100% mortality by 300 days p.i., but spontaneous mortality of uninfected STZ-NA mice was 40%. Dyslipidemia did not develop until 6 months after STZ-NA treatment and plasma insulin rose progressively over 6 months. In contrast, beta cell mass and function progressively decline in human type 2 DM; a process that may begin more than a decade before diagnosis[42]. Elevated IL-6 in H37Rv-infected STZ-NA mice was also reported by Wang et al.[40], but in their study this occurred by day 30 p.i., at a time point when Cheekatla et al.[39] found higher IL-6 levels in euglycemic than in STZ-NA mice. Hyperinflammation in the Wang study was attributed to increased STAT3 phosphorylation and NFAT5 activation. Finally, Alim et al.[41] reported significantly lower *M. bovis* BCG-stimulated IL-6 production by AMs and peritoneal macrophages from diabetic mice after 30 weeks on EDD, compared to euglycemic controls following ex vivo challenge.

These disparate results likely reflect model-specific conditions and highlight the need for additional research.

In our study, IL-6 levels in lung were no higher in HFD-STZ mice than the other infected groups. We found a CD11c⁺CD11b⁺Ly6C^lo MHCII⁺ myeloid cell population, probably myeloid DC and activated macrophages, that was exclusively increased in HFD-STZ mice after infection. We also found increased number of TUNEL⁺ cells in the lesions and higher IL-1α and IL-1β levels in lungs from HFD-STZ mice. These potentially linked phenomena may underlie the augmented immune pathology. Higher blood glucose levels in HFD-STZ vs STZ mice at the time of infection might have acutely exacerbated TB susceptibility between groups (Figs. 1, 2), but glucose levels were similar at later time points p.i. (Fig. 1d). Furthermore, the comparable susceptibility of CD and HFD-STZ mice infected with *glpK*-deficient *Mtb* H37Rv argues against that possibility. Low BMI is a human TB susceptibility factor[43],

but body weight prior to infection was not different between STZ and HFD-STZ mice (Fig. 1b). The protective effect of high BMI in human TB[44] was not exhibited by HFD mice in our study.

Loss of adipose tissue has wide ranging physiological consequences, as it is an endocrine tissue that regulates physiological processes of metabolism, energy balance and interacts with innate and adaptive immune responses[45,46]. In the HFD-STZ mouse model of type 2 DM, we found a marked reduction in adipocyte size and tissue mass attributable to lower de novo lipogenesis and increased lipolytic flux. This metabolic perturbation may contribute to wasting in TB disease and, importantly, was associated with elevated plasma levels of glycerol and FFA characteristic of type 2 DM. Glycerol utilization is not limiting in vivo for *Mtb* replication or immune pathology in C57BL/6J mice with normal glucose metabolism[27]. We found that *glpK*-deficient *Mtb* had a reduced ability to exploit the HFD-STZ host environment, and this TB phenotype was restored by *glpK* complementation. Increased TB severity in diabetic hosts has been attributed to the biochemical pathways of diabetic complications driven by chronic hyperglycemia[31]. Our data identified elevated plasma glycerol as an additional factor contributing to TB severity in a mouse model of type 2 DM. We attributed the reversal of glycerol elevation in plasma by 8 weeks (Fig. 4d) to adipose tissue wasting and utilization by bacteria or other tissues. Nonetheless, glycerol elevation early in the course onset of infection was sufficient to adversely impact host defense. Interestingly, feeding glycerol to STZ mice increased the lung bacterial load and lesion area (Fig. 5h, i). Glycerol has not been measured in clinical TB studies to our knowledge. Our findings raise this as a question that should be addressed in future clinical research.

In summary, here we present evidence linking TB severity in type 2 DM to elevated systemic levels of glycerol, uptake of which by *Mtb* associated with its virulence. Our results do not exclude the possibility that the combined metabolic disorders in HFD-STZ mice can also amplify diabetic immunopathy through established complication pathways such as RAGE signaling, dysfunctional protein kinase C activation, polyol or hexosamine pathway flux, oxidative stress and/or epigenetic reprogramming[34,47]. While the clinical implications of glycerol utilization by *Mtb* in human TB-DM comorbidity remain to be explored, the ability of metformin, a leading host-directed therapy candidate for TB[48], to attenuate glycerol release from adipocytes[49] supports the therapeutic relevance of targeting glycerol metabolism.

## Methods

### Mice
Age matched (6–8 weeks old) male C57BL/6J mice obtained from Jackson Laboratory (Bar Harbor, ME) were used for all the experiments and were housed in the Animal Medicine facility at UMCMS where experiments were performed under protocols approved by the Institutional Animal Care and Use Committee (protocol 202100197) and Institutional Biosafety Committee (protocol I-161). Mice were maintained at controlled temperature (23 °C ± 1), 12 h dark/light cycle, and at 50% ± 20 humidity with free access to food and water. Mice in the control diet (CD) group were fed Prolab Isopro RMH 3000 chow (14% kcal % fat; LabDiet, St. Louis, MO), while those in the streptozotocin (STZ) group were fed the CD and treated with a single i.p. dose of 150 mg/kg STZ (Sigma-Aldrich) 4 weeks prior to *Mtb* infection. Mice in the high fat diet group (HFD) were fed a 60% kcal % fat diet (Research Diets, Inc., New Brunswick, NJ #D12492) for 8 weeks prior to infection, while mice in the HFD-STZ group were treated with STZ 4 weeks after initiating the HFD. Mice in glycerol water (GW) were given a 5% glycerol solution in water ad libitum 8 weeks before the infection and 4 weeks before treatment with STZ. Mice were kept on the diet or the special waters for the entire duration of the experiment. Blood glucose levels were measured 10 days after STZ injection and the mice with levels

>300 mg/dl were included in the study. Mice were euthanized by asphyxiation using a chamber with carbon dioxide and cervical dislocation as a secondary method.

### *Mtb* infection
Eight weeks after starting on the respective diets, mice were aerosol infected with *Mtb* Erdman, wild type H37Rv or H37Rv *glpK* mutant (*ΔglpK*) or a complemented strain (Glpk-Comp) using a Glas-Col Inhalation Exposure System (Terre Haute, IN) set to deliver ~100 CFU to the lung. H37Rv and the *ΔglpK* mutants were shown to produce similar levels of PDIM by thin layer chromatography. Delivery dose was confirmed by sacrificing 4–5 mice one day after infection. Mice were weighed weekly, blood glucose measurements were performed with a BD Logic glucometer (Becton Dickinson, Franklin Lakes, NJ) biweekly and blood samples and tissues were taken at specified times after infection. Standard BSL-3 and ABSL-3 operating procedures were followed, using appropriate personal protective equipment, double containment to move samples between facilities, and Vesphene III (STERIS, Mentor, OH) for decontamination.

### Metabolic analysis
The insulin tolerance test (ITT) and glucose tolerance test (GTT) were performed after 6 h or overnight fasting, respectively. Mice were injected i.p. with 0.75 U/kg insulin for ITT or 1 mg/kg glucose for GTT and blood glucose was measured before and at several time points after injection.

### Lung bacterial burden
At specified time p.i., left lung lobes from *Mtb*-infected mice were homogenized in PBS containing 0.05% Tween 80. Serial dilutions of homogenates were plated on 7H11 agar and colonies were counted after 3 weeks.

### Flow cytometry
Cells isolated from tissues were stained with Zombie Aqua Viability Kit and for CD45 (30-F11), CD3 PE-Cy7 (17A2), CD11c PE (N418), CD11b PercP-Cy5.5 (M1/70), Ly6C A700 (HK1.4), Ly6G FITC (1A8) and MHC-II Bv421 (M5/114.15.2) (Biolegend, San Diego, CA). Another panel was used to identify activated T cells in lymph nodes and lungs: CD45 APC-Cy7 (30-F11), CD4 FITC (GK1.5), CD8 PE (YTS156.77), CD3 PercP-Cy5.5 (17A2), TNF-α APC (MP6-XT22), Ly6C A700 (HK1.4) and IFN-γ Bv421 (XMG1.2). All antibodies were used in a 1:200 dilution, except for MHC-II that was 1:50 dilution. Antibodies were validated by Biolegend in mouse splenocytes and extensive literature has been published using these antibodies. Data was acquired on an LSR-II (BD Biosciences, San Jose, CA) using DIVA software and was analyzed with FlowJo v10.8.1. The gating strategy for myeloid cell compartment is detailed in Fig. S2d.

### ELISA
Lung homogenates were used to measure IFN-γ, TNF-α, IL-6, IL-1α, and IL-β by ELISA following the manufacturer's protocol (R&D Systems, Minneapolis, MN). Lung homogenates were diluted 1/10 for IFN-γ, TNF-α, IL-1α and IL-β, and 1/2 for IL-6. Concentrations were back-calculated to the total volume of lung homogenate.

### Quantitative PCR
Zymo RNA extraction kit (Zymo Research Corp, Irvine, CA) was used to isolate total RNA from tissues or cells. Equal amounts of RNA were retro-transcribed with High Capacity Reverse Transcription Kit (Applied Biosystems). Real-time PCR for *MARCO* and *CD14* was performed at 60 °C annealing temperature using Applied Biosystems SYBR Green PCR mix. *TPB* and *GAPDH* were used as housekeeping genes and primer sequences were designed by Primer3 input. Primer sequences were: *MARCO* Fw 5-gaagacttcttgggcagcac-3'; Rv 5'-gtgagcaggatcaggtggat-3'; *CD14* Fw 5'-gtcaggaactctggctttgc-3' and Rv

5'-ggctttacccactgaacc-3'; *TPB* Fw 5'-gaagctgcggtacaattccag-3' and Rv 5'-cccccttgtacccttcaccaat-3'; *GAPDH* Fw 5'-acccagaagactgtggatgg-3'; and Rv 5'-cacattgggggtaggaacac-3'. Results were calculated as fold change to the control group by the delta-delta Ct method.

### Morphological analysis

Images of pgWAT were taken on a bright field microscope at 4X magnification. Using ImageJ 1.46r (NIH, Bethesda, MD), cell area was quantified for pgWAT in ~200 cells. Lung lesion area was measured in proportion to total lung area in images from H&E stained lungs using ImageJ 1.46r, pictures were taken on a bright field microscope at 2X magnification. The whole surface area of the multilobe half lung for each mouse was quantified. For oil red O staining, lipid positive area was measured in proportion to the total lung area. TUNEL staining was quantified by counting number of TUNEL⁺ cells in proportion of the total number of cells in the tissue section.

### Nuclei imaging in T cells

T cells were isolated from spleens from control or HFD-STZ mice using Dynabeads™ Untouched™ Mouse T Cells Kit (Invitrogen) and following the manufacturer's guidelines. T cells were fixed with 4% paraformaldehyde and spun down onto a glass slide. Cells were mounted using mounting media with DAPI (Molecular Probes) and they were imaged using a fluorescent microscope with 10X magnification. Nucleus diameter was measured using ImageJ 1.46r.

### Biochemical analysis

Glucose (Sigma-Aldrich), glycerol (Sigma-Aldrich), FFA (Cell Biolabs, Inc., San Diego, CA) and insulin (R&D Systems) were analyzed in plasma or tissue homogenate following the manufacturer's protocols.

### Immunoblotting

Total adipose tissue protein was quantified by BCA assay (Pierce Biotechnology, Rockford, IL) and SDS-PAGE separated. Fifteen μg of protein were used to measure ATGL, 10 μg protein for FASN and ACLY and 5 μg protein for phosphor-HSL, HSL and β-actin. Antibodies rabbit anti-mouse (Cell Signaling Technology Inc., Danvers, MA) phospho-ACLY (#4331), ACLY (#4332), FASN (#3189), ATGL (#2138), phospho-HSL (#4137), and total HSL (#4107) were used at 1:1000 dilution and β-actin (#4970) was used at 1:10,000 dilution. Bands were quantified by ImageJ 1.46r. The quantifications of the proteins of interest were calculated related to β-actin that was probed on the same gel or run in parallel in the case of ATGL. Validation of these antibodies was done by Cell Signaling using mouse cell lines and extensive literature has been published with these antibodies.

### Statistical analysis

Data was collected using Microsoft Excel for Microsoft 365 MSO (Version 2304 Build 16.0.16327.20200) 64-bit. GraphPad Prism 7.04 was used for statistical analysis. When normality was confirmed and 2 groups were compared, data were analyzed using a Student's *t*-test or by the Mann-Whitney U test if otherwise. If more than 2 groups were compared among the same time point or type of strain One-Way ANOVA was used. For survival curves, the log-rank Mantel-Cox test was used. A *p*-value of $< 0.05$ was considered statistically significant.

### Study approval

The study was approved by the Institutional Biosafety Committee (IBC) and Institutional Animal Care and Use Committee (IACUC) of the University of Massachusetts Medical School, USA.

### Reporting summary

Further information on research design is available in the Nature Portfolio Reporting Summary linked to this article.

## Data availability

All data needed to reproduce this work can be found in the manuscript, figures and supplementary data. Source data are provided with this paper.

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

## Acknowledgements

This research was supported by NIH Grant R01HL081149 (H.K.), R01HL152078 (H.K. & A.S.), and NMRC/OFIRG/may-0096 (A.S.). We acknowledge Dr. David Guertin from UMCMS for sharing his graphic created with BioRender.com.

## Author contributions

N.M. and H.K. conceived the idea and designed the study. N.M., L.S., M.J., K.C., J.W., and K.W. performed the experiments. C.Smith, M.B., and C.Sassetti provided key resources and related information. N.M., H.K., and A.S. analyzed data and developed figures. H.K. oversaw the study. N.M., H.K., and A.S. wrote the manuscript. All authors discussed the results and commented on the manuscript.

## Competing interests

The authors declare no competing interests.
