## [Peer review file · Nature Communications]

REVIEWER COMMENTS

Reviewer #1 (Remarks to the Author):

The goal of this study was to gain insight into the mechanism(s) involved in the increase in tuberculosis susceptibility in type 2 diabetic patients. To this end, the authors used chow diet (CD) and high fat diet (HFD) fed mice injected or not with streptozotocin (STZ) to mimic type 1 diabetes (CD-STZ) and type 2 diabetes (HFD-STZ) and infected or not with mycobacterium tuberculosis (Mtb) and performed several in vivo tests (IPGTT, IPITT).

In all these models, they determined in the Mtb infected mice survival, lung bacteria load and lesions, macrophage and T cell populations in lung and lung draining lymph nodes. They measured in perigonadal WAT (pgWAT) from uninfected and Mtb-infected mice the fat mass, adipocyte size, triglyceride (TG) content and plasma glycerol and free fatty acid (FFA). Only in uninfected mice, they evaluated pgWAT basal and insulin-stimulated lipogenesis and lipolysis by measurement of key enzymes involved in these metabolic processes. Finally, in the HFD/STZ model, they compared the effect of wild type Mtb vs a glycerol kinase Mtb mutant on mouse survival, body weight, lung bacterial load and lung lesion. The authors observed that the uninfected and Mtb infected HFD/STZ mice have a very similar phenotype: hyperglycemic, hypoinsulinemic, glucose intolerant and slightly insulin-resistant. However, Mtb-infected HFD-STZ mice have a reduced survival compared to the uninfected HFD-STZ and to the Mtb-infected- CD, -STZ and -HFD mice. Lung from Mtb-infected HFD-STZ mice showed an increase in Oil-red staining, bacteria load, lesion and dendritic cells, macrophages and lymphocytes compared to the 3 others infected mouse models indicating an increase in lung immune pathology. This was associated with a significant increase in IL1beta and IL1alpha in the lung and a decrease in activated CD4+ and CD8+ T cells in lung-draining lymph nodes from infected HFD-STZ mice suggesting a perturbation in innate and adaptive immune responses. They observed that both uninfected and Mtb-infected HFD/STZ mice have a decrease perigonadal fat mass and adipocytes compared to HFD mice associated with a reduced TG content and a higher FFA and glycerol plasma levels but only in uninfected HFD/STZ mice they measured a decreased lipogenesis in pgWAT extract. Finally, they showed that glycerol kinase Mtb mutant infected HFD/STZ mice vs wild type Mtb infected HFD/STZ mice is less virulent and is associated with a better survival, no reduction in body weight, reduced lung bacteria burden and lesions. However, these effects of this mutant on lung were not observed in the Mtb mutant infected euglycemic chow diet fed mice. The authors conclude that high glycerol levels in T2D mice contribute to poor prognosis following infections as bacteria feed in part on glycerol.

General comments:

The rationale of the work is quite interesting as it is related to the questions as to how type 2 diabetic patients are more at risk to develop tuberculosis disease. The authors have demonstrated in previous published work the higher virulence of mycobacterium tuberculosis in the context of chronic

hyperglycemia (STZ model) due to delayed innate and adaptive immune responses in the lung. In the present study, the authors studied the development of TB disease in 3 different mouse models: HFD/obesity, STZ/type 1 diabetes and HFD-STZ/type 2 diabetes with a particular focus on the HFD/STZ model. In the HFD/STZ model, TB disease is more pronounced than in any of the other models and the authors suggested that this is mainly due to the increase in systemic glycerol availability as a carbon source for Mtb growth and survival in the HFD/STZ model.

Although potentially very interesting in term of theme of research this paper suffers from major weakness and in some instances caveats and the conclusions are weakly supported by the current data. The main reasons are the following and may provide useful comments to the authors for a very solid paper.

- 1) The HFD-STZ is claimed to be a T2D model. In fact, it is a T1D model plus HFD. It is the same as the authors published before (same dose STZ) but her HFD is added to the model. The model may not be relevant to T2D as the glycemia is dramatically high chronically (about 25mM). It may although be a late diabetes type 2 uncontrolled with major beta cell mass reduction. This could be interesting, but the authors should better defend this. The caveat is that before they said that this model without HFD is a T1D. We have little idea of what these mice are in term of diabetes. There is no measurements of beta cell mass and pancreatic insulin content.
- 2) The stats are often done using inappropriate test and, on many figures, we see minor changes that a scientist working in the field of diabetes would see immediately that there is no effect yet here in some figures some changes are claimed significant. For example, in Fig 1 H the ITT is nor different yet claimed to be. They use t test or Mann-Whitney when they should use ANOVA in many figures.
- 3) The glycerol hypothesis is interesting but at this stage weakly supported. Key additional direct support should be found like feeding mice with glycerol or additional experiments listed below.
- 4) The authors did not observe in MTB-infected HFD/STZ mice an increase in systemic glycerol level (Fig4d) and also, they did not demonstrate any increase in adipose tissue lipolysis in this model.
- 5) Also, several conclusions are based on observation done only with uninfected animals or without comparison in particular to the HFD models.

Major additional comments

- 1- The authors should repeat the experiment with glycerol kinase Mtb mutant and WT in the 3 different models (STZ, HFD and HFD/STZ) and measure at different times post-infection glycerol levels and evaluated at the end as described in fig 5, lung bacterial load, lesion and performed an oil red staining of lung sections.
- 2- To demonstrate the importance of adipose tissue lipolysis as a source of glycerol in the HFD/STZ model vs the HFD model they should measure lipolysis as performed in SupFig4d in pgWAT extracts from these two mouse models.

3- The authors mentioned in the title of Fig2 that 'combined dyslipidemia and hyperglycemia increase TB severity', however they did not present any data related to dyslipidemia and so they blood lipid so they should measure total cholesterol and triglycerides in the different mouse models

4- The authors should also evaluate the innate and adaptive immune response (as figure 3b) in lung from Mtb-infected HFD fed mice based on the recent paper from Albornoz et al (Cells, 2021) showing that obesity also induced pulmonary inflammation in a similar HFD model.

5- The authors should measure not only in uninfected HFD-STZ mice (Fig 3a) but also in Mtb-infected HFD-STZ mice Marco and Cd14 expression level in alveolar macrophages as indicator of recognition and phagocytosis of Mtb.

Minor

1- All the graphs should be present with the individual values to have a better idea of the distribution.

2- For the IPITT data (Fig 1H and Sup 1e), the authors should also perform the AUC over the 120 min period.

3- The authors should discuss why only in HFD mice they observed a reduction in body weight over diet weeks (comparison fig 1g and sup 1a).

4- The authors should show the insulin level during IPGTT.

5- Please indicate when was taken the first glycemia after STZ administration in Fig 1b and sup1d.

6- The TUNEL data in Fig1f should be quantified.

7- The authors should quantify the oil red staining data in Fig 4e.

Reviewer #2 (Remarks to the Author):

Reviewer Comments to Author

Martinez and colleagues provided interesting and important results on TB susceptibility in the T2DM mouse model than T1DM. The manuscript contributes towards the mechanism that drives susceptibility to TB, which is of general interest and value to the field. I have some major concerns below.

Major comments

1. What was the influence of HFD/STZ alone or in combination on the total cell numbers harvested from the lungs at 8wpi, this is important to add in fig. 2.
2. Fig 2c, authors should comment on possible reasons underlying why the effect on bacterial burdens was only seen in the lungs given circulating lipids (glycerol and FFA) should be in spleen and liver.
3. #cells on the y-axis of Fig 2d, Fig 3 and FigS3 is back-calculated from the % cells of the number of live cells acquired on the flow or total cells obtained from the entire lung? They should be back-calculated to total cells obtained from the lungs of each group of animals.
4. Fig3a, MARCO and CD14 mRNA expression from infected mice also need to be incorporated.
5. Fig 4e, add the quantification of the Oil red staining in the lungs and Fig 2f, add quantification of terminal deoxynucleotidyl transferase dUTP nick end labelling lung sections.
6. Fig 5g. authors should explain why they don't see the difference in lung pathology when infected with glpk mutant Mtb yet a significant increase in CFU
7. The manuscript showed mechanisms such as one; reduced MARCO/CD14 expression and CD4/CD8 T cells activation for delayed immune recognition. Two; Pre-activation of T cells in STZ-HFD group determined by nuclei >6uM diameter appear weak. Both of these mechanisms were was expected from previously published reports (ref 5-7) however, the magnitude of these differences are on the order of two unlikely driving the phenotype alone. This may not be particularly novel however rather inferred given the results were expected from previous studies. Importantly, for both 3a and 3c, it is critical to include Mtb infected groups. The infection is likely to influence the expression of MARCO/CD14 and RAGE-dependent preactivated stage. MARCO and CD14 expression in alveolar macrophages was reported by the group to cause delay and hyperactive immune response consisting of Th1, Th2 and Th17 cytokines. Therefore, I believe that data should be switched between Fig 3 and Fig S4. Three; reduced lipogenesis and increased lipolytic flux. However, there is no significance in the figure reflecting results on lipolysis. They may have to tone down this interpretation results only show similar or tend to decrease from the control group
8. What caused the muscle mass to reduce (but not the liver) in mice treated with STZ alone but not in HFD-STZ mice.
9. Fig S2b, could the authors explain why the lung CFU for each group is similar in both acute (4wpi) and chronic (20wpi) infections, there not much increase in burdens from week 4 to 20?
10. Authors should show the data for glpK mutant complemented with wild-type gene to ensure the phenotype is indeed reversed.
11. It is counterintuitive delayed innate activation and preactivated (though nonspecific) T cells contribute to susceptibility. Should preactivated T cells not control TB better?

Methods

1. All the experiments were performed in male mice, is there any reason for such sex bias. It appears that animals were kept of HFD (Fig 1a) during the entire experiments, however, is not clear from the method section.
2. Metabolic analysis regarding measuring insulin need to be added along with blood glucose alone.
3. Flow cytometry antibodies need to be added with information on clones?
4. Add more information in ELISA, what was the dilution of lung homogenates used to measure cytokines. The data was back-calculated to the amount of protein in the lung sample by performing a BCA assay? In that case, should be represented per mg rather than per ml.
5. Add details on how lung lesions were measured and acquired on what microscope and how it was quantified.
6. Immunoblotting, how much protein was loaded for SDS-PAGE is missing, also text showing the rows in the image (Fig S4c) looks weird/overlapping image.
7. Figure legend for Fig 5, add information on dose and mode of Mtb infection.
8. Add titles and time points at which the analysis was done on all graphs for consistency amongst figures throughout. At the moment some have such info and some don't.
9. The last paragraph of the results, Fig 1a should be corrected to Fig 2a.
10. Was the approximate 100 CFU dose confirmed by sacrificing mice 1 dpi to investigate uptake?
11. Was there a reason for using the Erdman strain for all initial experiments instead of H37Rv, considering the mutant is on an H37Rv background?
12. It seems as though for all the mutant experiments, 10 mice were used whereas 5 mice were used in H37Rv experiments. Is there a reason and could the authors clarify this in the Figure Legend since it states n=5-6?
13. Mortality graphs: Could the authors specify the sample number.
14. Could the authors provide a clearer justification as to why the different Mtb infection time points (or collection of data at the indicated time points): 14dpi, 6wpi, 8wpi, 20wpi and 24wpi.
15. There is no mention in the methods of ethics protocol approved numbers for animal work or standard operating procedures for TB work in a BSL3.
16. Nuclei imaging to be listed in the methods.

Reviewer #3 (Remarks to the Author):

The manuscript entitled “glycerol contributes to tuberculosis susceptibility in mice with type 2 diabetes” by Martinez et al. is a carefully developed study examining tuberculosis in mice with diabetes. The authors developed 2 independent models, a Type I model using the drug streptozotocin (STZ), and a Type II model induced with STZ and a high-fat diet. The authors clearly show that TB is worsened in mice with diabetes and confirm their previous work showing immune dysfunction in mice with diabetes. A surprising result of these studies was the discovery that glycerol in the blood enhances the growth of Mtb, contributing to its pathogenesis. This work is convincingly demonstrated by using an Mtb mutant deficient for glycerol metabolism. This strain did not exhibit the increased pathogenesis in the diabetes model. The study will become a well-used model for future studies to develop therapies to enhance the treatment of people with TB and diabetes. The paper is well referenced, well-written. There are two minor comments:

- 1.) The sentence in the abstract that reads: “Mice were fed a high fat diet (HFD) or treated with streptozotocin (STZ; type 1 DM model) or received both treatments (HFD-STZ; type 2 DM model), before aerosol Mycobacterium tuberculosis (Mtb) challenge.” Could be reworded to be more clear.
- 2.) Did the authors check if their mutant, and the parents strain are PDIM positive? Wang Q, et al, Science 2020 doi: 10.1126/science.aav5912 demonstrates that PDIM loss effects glycerol uptake. Domenech P, Reed MB. Microbiology (Reading). 2009 The M. tuberculosis strain H37rv is shown to lose PDIM biosynthesis when cultured in 7H9 in vitro.

We appreciate the comments and suggestions of the reviewers that have helped us produce a much enhanced manuscript. We have addressed the reviewers' comments and included significant changes in the new version of the manuscript that are indicated with red font in the markup version of the main text file.

Reviewer #1

1. The HFD-STZ is claimed to be a T2D model. In fact, it is a T1D model plus HFD. It is the same as the authors published before (same dose STZ) but her HFD is added to the model. The model may not be relevant to T2D as the glycemia is dramatically high chronically (about 25mM). It may although be a late diabetes type 2 uncontrolled with major beta cell mass reduction. This could be interesting, but the authors should better defend this. The caveat is that before they said that this model without HFD is a T1D. We have little idea of what these mice are in term of diabetes. There is no measurements of beta cell mass and pancreatic insulin content.

HFD plus STZ models the condition of advanced type 2 diabetes, where beta cell loss drives the progression from relative to absolute insulin deficiency in the insulin-resistant host. This model, which is designed to investigate the effects of the diabetic host environment rather than the pathogenesis of type 2 diabetes, is established in the literature (1-6). We agree with the reviewer that glucose concentrations in our studies were very high, but ketoacidosis was not detected, and mice showed normal activity indicating no neurologic impairment suggesting hyperosmolarity. In India, which has the largest population of people with TB-diabetes comorbidity, we reported that a median HbA1c of 10.5% in cohort of 256 adults with type 2 diabetes and pulmonary TB, which indicates uncontrolled hyperglycemia (7). Extreme elevation of blood sugar in the mouse model accelerates pathological glycation, which is a primary driver of diabetic complications. This facilitates experiments within a timespan where aging does not present an additional confounding factor. A similar rationale has been applied to the use of high cholesterol diet and genetic manipulations produce extreme cholesterol elevation in mouse models of atherosclerosis. A recent Pubmed search of <mouse and streptozotocin> retrieved 10,193 hits so its effects are well known. There was a not a compelling reason for us to attempt measuring beta cell mass in the pancreas of STZ-treated mice where these cells are largely depleted. As recommended by the reviewer, we have added text to the manuscript in defense of the model.

2. The stats are often done using inappropriate test and, on many figures, we see minor changes that a scientist working in the field of diabetes would see immediately that there is no effect yet here in some figures some changes are claimed significant. For example, in Fig 1 H the ITT is nor different yet claimed to be. They use t test or Mann-Whitney when they should use ANOVA in many figures.

We have revised all the statistical methods used in the manuscript and clarified this in each figure legend. We are aware that the changes in the ITT curve were small. While they were statistically significant, we agree with the reviewer that they might not be biologically significant. The text in the first subsection of Results has been revised to reflect this point.

3. The glycerol hypothesis is interesting but at this stage weakly supported. Key additional direct support should be found like feeding mice with glycerol or additional experiments listed below.

We performed new experiments that strongly support our initial conclusions. Mice fed standard chow were treated with glycerol in drinking water and treated or not with STZ. Mice that received the

combination of glycerol plus STZ had increased CFU and lung pathology, compared to control mice and mice only given glycerol or STZ (Fig. 5g-i).

4. The authors did not observe in MTB-infected HFD/STZ mice an increase in systemic glycerol level (Fig4d) and also, they did not demonstrate any increase in adipose tissue lipolysis in this model.

The only known source of glycerol, apart from the diet, is adipose tissue lipolysis. Accordingly, the loss of adipose tissue mass we identified when mice are treated with STZ and became insulin-deficient surely indicates that lipolysis was activated. Moreover, we measured elevation of plasma glycerol and FFA in these mice, which would only be plausibly explained by lipolysis of stored triglycerides. We measured glycerol plasma levels in mice infected with Mtb H37Rv and the glpk mutant strain and in this case, it took more than 8 weeks to bring glycerol levels down to control group. Also, the HFD/STZ mice had reduced levels of glycerol as infection was progressing, which reflected the loss of adipose tissue mass over time. We hypothesize that when systemic glycerol levels are high, Mtb can use it as a preferred carbon source. In the host environment of STZ-induced hyperglycemia, this exacerbates TB severity. The effect likely waned as glycerol levels declined, but glycerol elevation during the early phase of infection was sufficient to worsen immune pathology. When lean mice were fed glycerol in drinking water, systemic levels were sustained and the adverse impact on TB defense was comparable to the HFD/STZ mice.

5. Also, several conclusions are based on observation done only with uninfected animals or without comparison in particular to the HFD models.

We have added more experiments performed in infected mice and including all the groups (Fig. 5a-e)

Major additional comments

1. The authors should repeat the experiment with glycerol kinase Mtb mutant and WT in the 3 different models (STZ, HFD and HFD/STZ) and measure at different times post-infection glycerol levels and evaluated at the end as described in fig 5, lung bacterial load, lesion and performed an oil red staining of lung sections.

We have performed these experiments and added them to the manuscript (Fig. 5a-e, Fig. S6b)

2. To demonstrate the importance of adipose tissue lipolysis as a source of glycerol in the HFD/STZ model vs the HFD model they should measure lipolysis as performed in SupFig4d in pgWAT extracts from these two mouse models.

In a biocontainment safety level 3 laboratory it has been challenging to measure lipolysis in the tissue ex vivo. The time of tissue transportation and tissue weight measure in a biocontained space blunted any difference that we could have seen. Since the source of glycerol is mainly the adipose tissue and the loss of tissue mass is so dramatic when mice are treated with STZ, it is reasonable to conclude that lipolysis stemming from insulin deficiency was increased in this group of mice.

3. The authors mentioned in the title of Fig2 that ‘combined dyslipidemia and hyperglycemia increase TB severity’, however they did not present any data related to dyslipidemia and so they blood lipid so they should measure total cholesterol and triglycerides in the different mouse models.

Levels of total cholesterol, HDL, LDL, and triglycerides have been added to the manuscript (Fig. S1f). The levels of total and HDL cholesterol were elevated in uninfected HFD and HFD/STZ mice, and baseline LDL

was elevated in HFD/STZ mice, indicating that these mice mirrored features of dyslipidemia seen in humans with diabetic and non-diabetic obesity. Trends for lower levels of some lipids by 8 weeks after Mtb infection in mice mirrored data we previously reported from a human cohort (8). We also previously reported that extreme elevation of cholesterol exacerbates TB (9, 10) but we do not believe that the moderate perturbation of cholesterol in current experiments exerted a detectable influence on the host-pathogen interaction.

4. The authors should also evaluate the innate and adaptive immune response (as figure 3b) in lung from Mtb-infected HFD fed mice based on the recent paper from Albornoz et al (Cells, 2021) showing that obesity also induced pulmonary inflammation in a similar HFD model.

The interesting results in the Albornoz study were not reproduced in our experiments (Fig. 2). While both studies used C57BL/6 mice, the diets were different and, importantly, Albornoz reported that increased susceptibility and IFN γ production were linked to gut microbiome which certainly differed from our mouse colony. We believe that the immunologic variables measured in our study are sufficient to support our key observations on the impact of glycerol. Our manuscript already comprises five main text figures and six supplemental figures with a combined total of 56 panels. A deeper investigation of the immune response and microbiome composition in HFD-STZ mice, while of potential interest, lies beyond the scope of the current manuscript.

5. The authors should measure not only in uninfected HFD-STZ mice (Fig 3a) but also in Mtb-infected HFD-STZ mice Marco and Cd14 expression level in alveolar macrophages as indicator of recognition and phagocytosis of Mtb.

We previously reported that in mice with chronic STZ-induced hyperglycemia, MARCO and CD14 are downregulated specifically on alveolar macrophages and not macrophages from other compartments (11). We proposed that this slowed the initial innate recognition of inhaled Mtb and thereby delayed priming of an adaptive immune response that was otherwise intact when fully expressed. Resident alveolar macrophages are the most numerous leukocytes in airways of uninfected mice, but they rapidly become a minority population after aerosol Mtb infection when recruited myeloid and lymphoid cells fill the airways and lung parenchyma (12). As suggested by the reviewer, we measured MARCO and CD14 gene expression in lung cells of mice 8 weeks after Mtb infection. As we predicted, the expression of MARCO and CD14 was similar in all the groups.

Minor Comments

1. All the graphs should be present with the individual values to have a better idea of the distribution.

This change has been made in the manuscript.

2. For the IPITT data (Fig 1H and Sup 1e), the authors should also perform the AUC over the 120 min period.

This graph has been added to the manuscript.

3. The authors should discuss why only in HFD mice they observed a reduction in body weight over diet weeks (comparison fig 1g and sup 1a).

A comment has been added to the manuscript (Page 5).

4. The authors should show the insulin level during IPGTT.

The production of insulin by the STZ and the HFD-STZ mouse groups is remarkably low, since we killed the beta cells in the pancreas. Given that hyperglycemia and tissue adipose wasting developed in mice, we are confident that insulin production was severely reduced.

5. Please indicate when was taken the first glycemia after STZ administration in Fig 1b and sup1d.

Blood glucose levels were measured 10 days after injecting the mice with STZ, this comment has been added to the manuscript in the methods section.

6. The TUNEL data in Fig1f should be quantified.

A graph with the quantification of TUNEL staining has been added to the manuscript (Fig. 3b).

7. The authors should quantify the oil red staining data in Fig 4e.

Quantification of the oil red-O staining has been added to new Fig. 3c.

Reviewer #2

1. What was the influence of HFD/STZ alone or in combination on the total cell numbers harvested from the lungs at 8 wpi, this is important to add in fig. 2.

Mice on a HFD or HFD-STZ treated mice had comparable number of total lung cells as the control group. Only the mice that were treated with STZ had a significant increase in the cell number compared to the control mice. This graph has been added to the manuscript (Fig. S2d).

2. Fig 2c, authors should comment on possible reasons underlying why the effect on bacterial burdens was only seen in the lungs given circulating lipids (glycerol and FFA) should be in spleen and liver.

While Mtb disseminates widely, the lung is a uniquely susceptible site of bacterial replication for reasons that are incompletely understood despite decades of research (13). Our data are compatible with that prior experience, as bacterial burden in the lungs exceeded that in liver or spleen by 15 to over 100-fold. We found that elevated glycerol exacerbated bacterial burden and inflammation in the lung only in

combination with hyperglycemia. Since it is known that the *glpK* mutant of H37Rv is fully virulent in normal mice (14), we concluded that the impact of excess glycerol fueling bacterial replication was insufficient to overcome normal host defense mechanisms to a detectable degree. Our results indicate that acute hyperglycemia and elevated glycerol have a synergistic adverse impact on host defense in the lung. We speculate that since normal hosts control *Mtb* in the liver and spleen more effectively than in the lung, the synergism between elevated glucose plus glycerol is insufficient to accelerate bacterial replication above the rate of elimination by the host.

3. #cells on the y-axis of Fig 2d, Fig 3 and FigS3 is back-calculated from the % cells of the number of live cells acquired on the flow or total cells obtained from the entire lung? They should be back-calculated to total cells obtained from the lungs of each group of animals.

The data shown in the figures was back-calculated to the total cells obtained from the lungs.

4. Fig3a, MARCO and CD14 mRNA expression from infected mice also need to be incorporated.

We measured the expression of MARCO and CD14 in all the groups in infected mice. However, since the percentage of alveolar macrophages is small compared to uninfected mice, we are not surprised to see no changes in the lung cells. For details, please see our response to major comment #5 from Reviewer 1.

5. Fig 4e, add the quantification of the Oil red staining in the lungs and Fig 2f, add quantification of terminal deoxynucleotidyl transferase dUTP nick end labelling lung sections.

As requested by the reviewer, we have added quantification to the revised manuscript (Fig. 3b,c).

*6. Fig 5g. authors should explain why they don't see the difference in lung pathology when infected with *glpK* mutant *Mtb* yet a significant increase in CFU*

In the mouse aerosol TB model, once lung CFU exceeds 10^6 the correlation between bacterial burden and immune pathology may not be seen, particularly at the 24-week timepoint of the experiment in Fig. 5. Our previous published data show that bacterial replication promotes immunostimulatory macrophage necrosis that in turn drives immune pathology (12). A higher plateau bacterial burden may persist for an extended period of time after bacterial replication is slowed by host immunity. In the current experiments, we show that glycerol increases pathology and strongly increases cell death (Fig. 2F) which we attribute to higher bacterial replication fueled by glycerol.

*7. The manuscript showed mechanisms such as one; reduced MARCO/CD14 expression and CD4/CD8 T cells activation for delayed immune recognition. Two; Pre-activation of T cells in STZ-HFD group determined by nuclei >6uM diameter appear weak. Both of these mechanisms were expected from previously published reports (ref 5-7) however, the magnitude of these differences are on the order of two unlikely driving the phenotype alone. This may not be particularly novel however rather inferred given the results were expected from previous studies. Importantly, for both 3a and 3c, it is critical to include *Mtb* infected groups. The infection is likely to influence the expression of MARCO/CD14 and RAGE-dependent preactivated stage. MARCO and CD14 expression in alveolar macrophages was reported by the group to cause delay and hyperactive immune response consisting of Th1, Th2 and Th17 cytokines. Therefore, I believe that data should be switched between Fig 3 and Fig S4. Three; reduced lipogenesis and increased lipolytic flux. However, there is no significance in the figure reflecting results on lipolysis. They may have to tone down this interpretation results only show similar or tend to decrease from the control group.*

Figures have been rearranged and changes in these issues have been included in the manuscript.

8. What caused the muscle mass to reduce (but not the liver) in mice treated with STZ alone but not in HFD-STZ mice.

We hypothesize that the constant input of fatty acids from the diet is delaying the development of sarcopenia resulting from insulin deficiency and glycosuria (15).

Impaired insulin signaling in the liver constrains insulin-stimulated glycogen synthesis from glucose, but lipid synthesis is exacerbated. In the skeletal muscle and during insulin resistance glucose gets diverted to the liver to be used for de novo lipogenesis increasing TG synthesis. This explains how the skeletal muscle has a reduced mass during insulin resistance, while liver maintains it. However, both the skeletal muscle and the liver might benefit from the dietary lipid and store it as TG keeping the mass for a longer period than the adipose tissue.

9. Fig S2b, could the authors explain why the lung CFU for each group is similar in both acute (4wpi) and chronic (20wpi) infections, there not much increase in burdens from week 4 to 20?

These two time points should not be compared together in this case, as they are 2 separate infections. However, the bacterial growth in the lung plateaus, depending on the initial dose and the bacterial strain this can happen at different time points. Our previous work on TB susceptibility in hyperglycemic showed that the major deficit is the delayed expression adaptive immunity, which once invoked is capable of restricting lung bacterial burden to a plateau value (16).

10. Authors should show the data for glpK mutant complemented with wild-type gene to ensure the phenotype is indeed reversed.

We performed new experiments including the glpk mutant complemented strain. Results are shown in Fig. 5c,d. The glpk mutant complement lung bacterial growth and pathology showed the same trend as the WT H37Rv strain.

11. It is counterintuitive delayed innate activation and preactivated (though nonspecific) T cells contribute to susceptibility. Should preactivated T cells not control TB better?

The impaired recognition of inhaled Mtb bacteria by resident alveolar macrophages slows the initial recruitment of innate immune responding cells including myeloid dendritic cells (DC) to the site of infection. Consequently, time required for DC to acquire Mtb bacilli and antigens from infected macrophages and to traffic to the lung draining lymph nodes is increased by several days. This delays the priming of Mtb antigen-specific CD4+ T cells and their subsequent trafficking to the lung. The innate response is unable to restrict Mtb replication, so lung bacterial burden increases logarithmically until adaptive immunity is expressed. The outcome is higher bacterial burden and more immune pathology in diabetic mice(11, 16, 17). The non antigen-specific T cell hyperresponsiveness that we described in uninfected mice with chronic hyperglycemia (18) might contribute to increased immune pathology but would not be expected to enhance host defense. Once diabetic mice mount an adaptive immune response to TB it fully capable of slowing Mtb replication and restricting lung bacterial burden to a plateau level, albeit higher than that in normoglycemic mice.

Methods

1. All the experiments were performed in male mice, is there any reason for such sex bias. It appears that

animals were kept of HFD (Fig 1a) during the entire experiments, however, is not clear from the method section.

We have used males to keep for consistency with our previous studies, as females do not respond to STZ as much as males. Typically, ~50% of the females need to be reinjected with STZ to show beta cell loss. Details about the duration of the HFD in the studies have been added to methodology.

2. Metabolic analysis regarding measuring insulin need to be added along with blood glucose alone.

Insulin levels were measured at the time of infection, 4 weeks after STZ treatment (Fig. 1f). However, because the STZ and the HFD-STZ mouse groups do not produce insulin, we did not measure it during GTT or ITT protocols.

3. Flow cytometry antibodies need to be added with information on clones?

This information has been added to the methods section.

4. Add more information in ELISA, what was the dilution of lung homogenates used to measure cytokines. The data was back-calculated to the amount of protein in the lung sample by performing a BCA assay? In that case, should be represented per mg rather than per ml.

The dilution of the samples for the ELISA has been added to the methods section. The data was back-calculated to the total volume of lung homogenate. This information has been added to the methods section.

5. Add details on how lung lesions were measured and acquired on what microscope and how it was quantified.

These details have been added to the methods section of the manuscript.

6. Immunoblotting, how much protein was loaded for SDS-PAGE is missing, also text showing the rows in the image (Fig S4c) looks weird/overlapping image.

This information has been added to the methods section of the manuscript.

7. Figure legend for Fig 5, add information on dose and mode of Mtb infection.

This information has been added to the manuscript.

8. Add titles and time points at which the analysis was done on all graphs for consistency amongst figures throughout. At the moment some have such info and some don't.

Figures have been changed and information added to them

9. The last paragraph of the results, Fig 1a should be corrected to Fig 2a.

This has been corrected.

10. Was the approximate 100 CFU dose confirmed by sacrificing mice 1 dpi to investigate uptake?

The infecting dose of Mtb was confirmed with 4-5 mice 24 hours post-infection. This information has been added to the methods section.

11. Was there a reason for using the Erdman strain for all initial experiments instead of H37Rv, considering the mutant is on an H37Rv background?

To study susceptibility and in long studies like we prefer using the most virulent strain of Mtb. At the beginning of this project, we did not consider using the glpk mutant which was only available in H37Rv. However, we also believe the use of two different virulent wildtype Mtb strains controls for potential strain-dependent effects.

12. It seems as though for all the mutant experiments, 10 mice were used whereas 5 mice were used in H37Rv experiments. Is there a reason and could the authors clarify this in the Figure Legend since it states n=5-6?

When we do survival studies, we include ~n=10, for other assays n=6. For the H37Rv mouse infected groups, there were some deaths and that is reflected as a smaller number some in the end of study measurements.

13. Mortality graphs: Could the authors specify the sample number.

This has been added to the figure legend.

14. Could the authors provide a clearer justification as to why the different Mtb infection time points (or collection of data at the indicated time points): 14dpi, 6wpi, 8wpi, 20wpi and 24wpi.

Early time points were used to study T cell priming in the lymph nodes, which happens between day 7 and day 14 after infection. Eight week time point, was the minimum time where we would see a good susceptibility for the different conditions and where the bacterial growth has already plateaued. Six weeks p.i. was used to do the GTT and ITT studies, to give the mice a 2 week time rest before the end-of-study harvest. Long time points were used for survival curves and chronic effect of the diet on adipose tissue, and for H37Rv strain, which is not as virulent. Relevant comments have been added to the manuscript.

15. There is no mention in the methods of ethics protocol approved numbers for animal work or standard operating procedures for TB work in a BSL3.

This has been added to the manuscript.

16. Nuclei imaging to be listed in the methods.

This has been included in the manuscript.

Reviewer #3

1. The sentence in the abstract that reads: "Mice were fed a high fat diet (HFD) or treated with streptozotocin (STZ; type 1 DM model) or received both treatments (HFD-STZ; type 2 DM model), before aerosol Mycobacterium tuberculosis (Mtb) challenge." Could be reworded to be more clear.

This has been rephrased in the new version of the manuscript.

2. Did the authors check if their mutant, and the parents strain are PDIM positive? Wang Q, et al, Science 2020 doi: 10.1126/science.aav5912 demonstrates that PDIM loss effects glycerol uptake. Domenech P, Reed MB. Microbiology (Reading). 2009 The M. tuberculosis strain H37rv is shown to lose PDIM biosynthesis when cultured in 7H9 in vitro.

This has been tested and both the H37Rv and the Δ glpK mutants were shown to produce similar levels of PDIM by TLC. Additionally, the complementation experiment ruled out a functionally significant difference between WT and the mutant.

References

1. Srinivasan K, Viswanad B, Asrat L, Kaul CL, Ramarao P. Combination of high-fat diet-fed and low-dose streptozotocin-treated rat: a model for type 2 diabetes and pharmacological screening. *Pharmacological research*. 2005;52(4):313-20.
2. Islam MS, Loots du T. Experimental rodent models of type 2 diabetes: a review. *Methods and findings in experimental and clinical pharmacology*. 2009;31(4):249-61.
3. Zhang M, Lv XY, Li J, Xu ZG, Chen L. The characterization of high-fat diet and multiple low-dose streptozotocin induced type 2 diabetes rat model. *Experimental diabetes research*. 2008;2008:704045.
4. Parilla JH, Willard JR, Barrow BM, Zraika S. A Mouse Model of Beta-Cell Dysfunction as Seen in Human Type 2 Diabetes. *Journal of diabetes research*. 2018;2018:6106051.
5. Magalhães DA, Kume WT, Correia FS, Queiroz TS, Allebrandt Neto EW, Santos MPD, et al. High-fat diet and streptozotocin in the induction of type 2 diabetes mellitus: a new proposal. *Anais da Academia Brasileira de Ciências*. 2019;91(1):e20180314.
6. Hossain MJ, Kendig MD, Letton ME, Morris MJ, Arnold R. Peripheral Neuropathy Phenotyping in Rat Models of Type 2 Diabetes Mellitus: Evaluating Uptake of the Neurodiab Guidelines and Identifying Future Directions. *Diabetes & metabolism journal*. 2022;46(2):198-221.
7. Kornfeld H, Sahukar SB, Procter-Gray E, Kumar NP, West K, Kane K, et al. Impact of Diabetes and Low Body Mass Index on Tuberculosis Treatment Outcomes. *Clinical infectious diseases : an official publication of the Infectious Diseases Society of America*. 2020.
8. Prada-Medina CA, Fukutani KF, Pavan KN, Gil-Santana L, Babu S, Lichtenstein F, et al. Systems Immunology of Diabetes-Tuberculosis Comorbidity Reveals Signatures of Disease Complications. *Sci Rep*. 2017;7(1):1999.
9. Martens GW, Arikan MC, Lee J, Ren F, Vallerskog T, Kornfeld H. Hypercholesterolemia impairs immunity to tuberculosis. *Infect Immun*. 2008;76(8):3464-72.
10. Martens GW, Vallerskog T, Kornfeld H. Hypercholesterolemic LDL receptor-deficient mice mount a neutrophilic response to tuberculosis despite the timely expression of protective immunity. *J Leukoc Biol*. 2012.
11. Martinez N, Ketheesan N, West K, Vallerskog T, Kornfeld H. Impaired Recognition of Mycobacterium tuberculosis by Alveolar Macrophages From Diabetic Mice. *J Infect Dis*. 2016;214(11):1629-37.
12. Repasy T, Lee J, Marino S, Martinez N, Kirschner DE, Hendricks G, et al. Intracellular Bacillary Burden Reflects a Burst Size for Mycobacterium tuberculosis In Vivo. *PLoS Pathog*. 2013;9(2):e1003190.
13. North RJ, Jung YJ. Immunity to tuberculosis. *Annu Rev Immunol*. 2004;22:599-623.
14. Bellerose MM, Proulx MK, Smith CM, Baker RE, Ioerger TR, Sasseti CM. Distinct Bacterial Pathways Influence the Efficacy of Antibiotics against Mycobacterium tuberculosis. *mSystems*. 2020;5(4).
15. Al Saedi A, Debruin DA, Hayes A, Hamrick M. Lipid metabolism in sarcopenia. *Bone*. 2022;164:116539.
16. Martens GW, Arikan MC, Lee J, Ren F, Greiner D, Kornfeld H. Tuberculosis susceptibility of diabetic mice. *Am J Respir Cell Mol Biol*. 2007;37(5):518-24.
17. Vallerskog T, Martens GW, Kornfeld H. Diabetic Mice Display a Delayed Adaptive Immune Response to Mycobacterium tuberculosis. *J Immunol*. 2010;184:6275-82.

18. Martinez N, Vallerskog T, West K, Nunes-Alves C, Lee J, Martens GW, et al. Chromatin decondensation and T cell hyperresponsiveness in diabetes-associated hyperglycemia. *J Immunol.* 2014;193(9):4457-68.

REVIEWERS' COMMENTS

Reviewer #1 (Remarks to the Author):

The authors have adequately addressed all the concerns I raised. I have no further comments at this point.

Reviewer #2 (Remarks to the Author):

I am happy with the author responses to my concerns raised

Reviewer #3 (Remarks to the Author):

The authors have masterfully answered all of the reviewers questions; this revised manuscript is excellent and much improved from earlier submission.